# Evaluation of Automated Object-Detection Algorithms for Koala Detection in Infrared Aerial Imagery

**DOI:** 10.3390/s24217048

**Published:** 2024-10-31

**Authors:** Laith A. H. Al-Shimaysawee, Anthony Finn, Delene Weber, Morgan F. Schebella, Russell S. A. Brinkworth

**Affiliations:** 1UniSA STEM, University of South Australia, Mawson Lakes, SA 5095, Australia; anthony.finn@unisa.edu.au (A.F.); delene.weber@unisa.edu.au (D.W.); morgan.schebella@unisa.edu.au (M.F.S.); 2College of Science and Engineering, Flinders University, Tonsley, SA 5042, Australia; russell.brinkworth@flinders.edu.au

**Keywords:** koala monitoring, occlusion analysis, canopy cover effects, insect vision, bio-inspired vision signal processing

## Abstract

Effective detection techniques are important for wildlife monitoring and conservation applications and are especially helpful for species that live in complex environments, such as arboreal animals like koalas (*Phascolarctos cinereus*). The implementation of infrared cameras and drones has demonstrated encouraging outcomes, regardless of whether the detection was performed by human observers or automated algorithms. In the case of koala detection in eucalyptus plantations, there is a risk to spotters during forestry operations. In addition, fatigue and tedium associated with the difficult and repetitive task of checking every tree means automated detection options are particularly desirable. However, obtaining high detection rates with minimal false alarms remains a challenging task, particularly when there is low contrast between the animals and their surroundings. Koalas are also small and often partially or fully occluded by canopy, tree stems, or branches, or the background is highly complex. Biologically inspired vision systems are known for their superior ability in suppressing clutter and enhancing the contrast of dim objects of interest against their surroundings. This paper introduces a biologically inspired detection algorithm to locate koalas in eucalyptus plantations and evaluates its performance against ten other detection techniques, including both image processing and neural-network-based approaches. The nature of koala occlusion by canopy cover in these plantations was also examined using a combination of simulated and real data. The results show that the biologically inspired approach significantly outperformed the competing neural-network- and computer-vision-based approaches by over 27%. The analysis of simulated and real data shows that koala occlusion by tree stems and canopy can have a significant impact on the potential detection of koalas, with koalas being fully occluded in up to 40% of images in which koalas were known to be present. Our analysis shows the koala’s heat signature is more likely to be occluded when it is close to the centre of the image (i.e., it is directly under a drone) and less likely to be occluded off the zenith. This has implications for flight considerations. This paper also describes a new accurate ground-truth dataset of aerial high-dynamic-range infrared imagery containing instances of koala heat signatures. This dataset is made publicly available to support the research community.

## 1. Introduction

The ability to accurately enumerate wildlife populations is fundamental to many conservation and natural resource management programs [1,2]. For instance, accurately determining a species population size is essential for drawing up effective plans to protect endangered species [3], monitoring the movement and behaviour of migratory animals [4], managing populations of invasive species within eradication programs [5,6], and optimising the protection of wildlife during natural disasters. However, detecting wildlife is a challenging task. Wildlife moves and is often camouflaged against its background [7], and the environmental background is often “cluttered”, consisting of a complex interplay of elements. Despite the difficulty of detection, land managers require accurate data to achieve their objectives without negatively impacting wildlife. However, traditional wildlife surveying methods, including diurnal searches, nocturnal spotlighting [8,9], detection dogs and radio or satellite collars, camera traps [9], hand-held thermal imagers [10], and crewed aerial surveys [11] all have limitations. In addition, they are often time-consuming, labour intensive, and expensive, and some are hazardous [1].

More recently, uncrewed aerial vehicles (UAVs) or drones have become commercially available and used in diverse applications. UAVs have been used in conjunction with sensors such as colour and thermal cameras to monitor forests and detect animals such as livestock [12,13], kangaroos [14], deer [15,16], fur seals [17], sea turtles [18], monkeys [19], and koalas [20,21]. In these studies, the presence of animals was determined manually, by visually examining recorded imagery. However, while drones and cameras offer a new way of monitoring wildlife, manually inspecting the imagery is time consuming, tedious, and expensive, especially when there is a large amount of imagery [22].

The use of UAVs to assist in the automated detection of wildlife is not new. They have been used to detect animals such as wild turkeys [23], rabbits and chickens [24,25], cows [26], white-tailed deer [27], hippopotamus [28], seals [29], greater kudus, gemsboks, and hartebeests [1,30]. A more detailed literature review on the automated detection of wildlife using UAVs can be found in [31].

Automated deep-learning techniques can identify and extract critical features from imagery to detect objects of interest [32,33,34,35]. Supervised deep-learning techniques have historically shown promising results in wildlife and object detection within aerial imagery [23,26,36,37,38]. However, these techniques have limitations. Firstly, they require large amounts of training data, which are not always available for specific objects of interest. In addition, sometimes the available training data are noisy and hence require using certain techniques to eliminate the noisy training labels before using them in training, as deep neural networks can easily overfit noisy labels, leading to significant degradation in the results [34]. Although fine-tuning and data-augmentation techniques have significantly reduced the amount of training data required [39], a relatively large amount of data is still needed. Secondly, as neural networks use contextual features (i.e., multiple neighbouring pixels) within the relevant image set, the training data have to have enough object features (i.e., occupy sufficient pixels) for the network to learn how to classify patterns of interest, not just the intended objects of interest in the dataset(s). The need to use object features for classification can be a significant problem in datasets where the object of interest is very small (occupying fewer than 10 pixels) and occlusion further reduces the visible size of the object. Consequently, comparative neural-network algorithms such as Faster RCNN [40] and YOLO [41] techniques may not be able to learn the key features of items of interest with such small signatures. They could therefore fail during the training stage. While results may improve for these techniques when the images are scaled up, processing times would also increase.

Other factors that have effects on the detection techniques include the type of camera used and its resolution. For example, in thermal images, small objects generally exhibit even fewer features than their visible (colour) counterparts. This is because thermal cameras typically have lower resolutions than visible cameras and contiguous pixels are often not as thermally distinct from one another as the information in a colour image. Additionally, the key features of small objects are typically lost while passing through the pooling and convolutional layers of a deep neural network. For instance, a 32 × 32 pixel object will be represented (at most) as 1 pixel after five processing steps by a pooling layer in the VGG16 network. This means it could easily be missed if processed using more layers [42]. Therefore, there is a need for techniques that can detect objects with a low number of features and do not require large amounts of training data to address the aforementioned issues. Many image-processing techniques have been inspired by biology to address a diversity of applications. Flying insects undertake complex tasks such as detecting and chasing objects in high clutter backgrounds under a wide variety of weather and environmental lighting conditions [43,44,45,46,47,48]. Despite the small size, weight, and power draw of an insect brain, and its limited number of neurons [47], these animals perform complex tasks easily in real time. This remarkable capability has encouraged scientists to study the vision systems of insects and develop models that are inspired by or directly mimic them [47,49,50,51]. These bio-inspired vision methods can outperform traditional image-processing techniques in complex environments [52], and many of these bio-inspired techniques do not require training or prior knowledge of a scene’s lighting conditions [52]. Therefore, one of the aims of this paper is to develop and test such a method for use in the field of wildlife detection and compare it to existing techniques. The impetus for this research was a need for the accurate detection of koalas to mitigate potential harm that could arise from forest operations. Koalas are an iconic Australian species, but unfortunately, due to a range of factors, including habitat fragmentation, disease, vehicle strikes, poor genetic diversity, and bushfires [53], there is a significant decline in populations. In 2022, the Australian government declared koalas as endangered across much of their range under the Environment Protection and Biodiversity Conservation Act 1999. While they are listed as stable in the location of the study, there has been a concerted effort across Australia to protect this species, not only in native forests but also when they establish themselves in forest plantations.

Unlike conventional object-detection techniques, detecting wildlife in complex forest environments is a challenge due to several factors, such as dense trees, various weather conditions, and the ability of animals to camouflage within the environment [54]. The use of colour imagery for koala detection is problematic as koalas are typically hidden by tree foliage and the species blends in with the surrounding, meaning colour contrast is low [55]. UAVs and thermal imaging technologies have been used widely in forestry and wildlife conservation [56]. Recently, several researchers have used UAVs and thermal cameras to detect koalas [7,39,55,57]. The results are promising when compared to traditional methods as they offer a comparable detection accuracy, in addition to improved safety and surveying speeds [39]. However, these studies were conducted under temporal constraints (early in the morning) and in cold weather [39,55,58], which ensured there was a stronger contrast between the koala, a warm-blooded animal, and the surrounding trees and foliage and consequently increased the likelihood of accurate estimates of the koala population. From a practical point of view, however, land managers, such as forest companies, desire the ability to accurately assess population numbers during any season, under all weather conditions, and at any time of the day [20,21]. Having such a technology will help land managers to achieve their objectives without compromising the welfare of wildlife in the area.

To enable a better understanding of the challenges associated with koala detection in infrared aerial imagery, the influence of occlusion by the canopy was examined using a combination of simulated and real data. The effect on the probability of detection of the relative geometry of the koalas and drone/camera was also analysed. This study aimed to investigate the effectiveness of automated object-detection systems for identifying koalas in eucalyptus plantations in southwest Victoria, Australia.

The main contributions of this paper are as follows:An accurate ground-truth dataset of aerial high-dynamic-range thermal imagery containing koalas in eucalyptus plantations. The dataset, known as Koala InfraRed Aerial Imagery (Kirai), has images from four flights at three different locations. The dataset is publicly available to the research community (https://github.com/LaithAlShimaysawee/KiraiDataset, accessed on 10 October 2024), acknowledging the shortage of such field data and the need for further research.The introduction of a new object-detection method and comparison to 10 existing state-of-the-art object-detection techniques for the detection of koalas in eucalyptus plantations.A pilot study on the effect of time of the day on automated koala detection in infrared aerial imagery and recommendations for future research.An analysis of the effect of occlusion by tree canopy on koala detection (and likely arboreal mammals in general) using a combination of simulated and real-world data.Recommendations regarding drone/camera angles on the probability of detection.

This paper is organised into two sections. Section 1 discusses the methodology of real-data collection, comparative detection techniques and their experimental settings, performance metrics, and the results of these detection techniques. Section 2 discusses the methodology and results of other experiments using real and simulation data to study the effects of several parameters on koala detection. These parameters are environmental, such as the temperature, tree canopy structure, and the koala’s position within the tree. Additionally, there are flight parameters, including factors related to the flight settings, such as the drone’s flight altitude and the camera’s depression angle.

## 2. Part 1: Evaluation of Several Detection Techniques on Koala Detection

### 2.1. Methodology of Part 1

This section describes the equipment used in the field trials and the recorded datasets. It also describes a benchmark of existing object-detection algorithms, the experimental settings, and the evaluation criteria used in the study.

#### 2.1.1. Camera and Drone

The model of infrared camera used was an ICI-8640 P-series (ICI, Beaumont, Manufacturers in Beaumont, TX, USA), which has a spectral band of 7–14 µm [59]. It had a pixel-depth (dynamic range) of 14 bits encoded in a 16-bit wrapper. The camera was attached to a bespoke payload, designed by researchers at the university of South Australia, which captured the high-dynamic-range (HDR) raw thermal images at a frame rate of 10 Hz (note, this has subsequently been upgraded to 30 Hz). The image resolution was 640 × 512 pixels. The focal length of the lens was 12.5 mm with manual focus. This translated to a field of view (FOV) of about 50° × 37.5°. The payload was mounted on a DJI Ronin Gimbal (DJI, Shenzhen, China) to provide spatial stability, carried by a DJI Matrice 600 drone (DJI, Shenzhen, China) [60].

#### 2.1.2. Survey Site and Real-Data Collection

Three datasets were recorded from two sites in southwestern Victoria. The first area was known to have a high population of koalas and comprised around three hectares (300 m × 100 m). Two flights were conducted at this site, one at 10:30 a.m. and one at 11:30 a.m., both on 14 November 2019. The data recorded from these flights are referred to as datasets A and B, respectively. Figure 1A,B shows two image samples from each dataset. The second site was around five hectares (300 m × 168 m). One flight was conducted at this site at 11:15 a.m. on 12 November 2019. The data recorded from this site is referred to as dataset C. Figure 1C shows two image samples from this dataset. A key reason for performing the flight missions later in the morning was to test the performance of the methods at a time that is more challenging for conventional infrared monitoring. The forest industry could be harvesting at any time of the day, and in all seasons, so it was important that some data in the warmer part of the day have been recorded. On both trial days, wind speed was around 20 km/h and ambient temperature was about 12–17 °C. It should be noted that the data were collected around the middle of November, which is considered the end of spring in Australia. The drone flew above the sites in a lawn-mower pattern at an altitude above the ground of approximately 60 m (35 m above the tree tops), with a constant forward speed of 8 m/s. The flight path was designed to have 50% minimum side overlap (at ground level) between images from adjacent transects. Datasets A, B, and C comprised 2770, 2530, and 4025 image frames, respectively. It took the drone between 8 and 15 min to cover the two sites. The location of each image was recorded by the drone using a real-time kinematic (RTK) carrier phase differential global position system (GPS).

To create ground truth, prior to conducting the trials, the trunk of every tree was marked with a unique identifier (ID) using chalk. Then, during the trial, for every tree trunk occupied by a koala, eight independent expert koala spotters individually identified the location of all koalas by tree ID, time of day, and GPS coordinates. On average, spotters took 1.5 ± 0.5 and 3.0 ± 1.13 h to survey both sites (mean ± standard deviation), respectively. Upon completion of all ground surveys, a list of every koala detected by every spotter was compiled, and the location of every found koala was independently verified through visual re-inspection of the sites using tree ID. It should be noted that, although unlikely, it is possible there were koalas not detected by either the eight ground observers or the drone-mounted sensor. The key purpose of pre-labelling all trees in a study area was to ensure the spotters findings remained independent, i.e., they did not share information about the koalas’ locations. In other words, whilst it would be more time-efficient for spotters to simply mark each tree where they spotted a koala, this would have prevented spotters from conducting each survey “blindly”. The second reason for labelling the trees was that finding the GPS location of trees in the sub-canopy is a notoriously challenging problem. The location uncertainty in such a task derives from many factors, including the attenuation and scattering of the GPS signal, the multi-path caused by the trees and ground, the poor geometry of the available satellites that are visible, and the increased jitter in the tracking loops (lower accuracy pseudo range measurements) caused by the lower signal-to-noise ratio of the GPS signal. In addition, spotters generally report the location at which they are standing (looking up at the koala) rather than a careful (vector offset) estimate of the centre of the tree or location beneath the koala.

The ground truth for the images from each dataset was found by first creating an orthomosaic image from the infrared imagery for each flight using Agisoft’s Metashape 2.1.3 software [61]. Then, using the GPS coordinates and tree IDs provided by the spotters as a guide, the koalas’ locations were manually marked on the orthomosaic. These locations were then manually adjusted to ensure all relevant aerial and ground-based observations corresponded with one another, noting that all koalas detected by ground-based spotters were clinging to a tree, i.e., no koalas were spotted on the ground. In other words, the locations of all detected koalas in the infrared imagery were extracted and the (x, y) pixel positions within each frame with a koala in it manually confirmed.

#### 2.1.3. Comparative Methods

A new object-detection method was introduced and compared to ten existing state-of-the-art object-detection techniques using three infrared datasets containing koalas in eucalyptus plantations.

The introduced technique is known as the Multiscale Object Bio-Inspired Vision Line Scanner (MOBIVLS) and draws inspiration from the visual pathways of flying insects. This vision system has many stages, each designed for different tasks, but in general the object-detection pipeline is composed of the photoreceptor cells (PRCs), lamina monopolar cells (LMCs), rectified transient cells (RTCs, and elementary small target motion detectors (ESTMDs) [47,49,62]. The photoreceptor cells are responsible for adapting to light changes. This adaptation helps to effectively compress high-dynamic-range images without losing important information. It also enhances the contrast between objects of interest and their surroundings using temporal processing to enhance object separation by up to 70% [63]. The lamina monopolar cells (LMCs) are responsible for removing spatial and temporal redundancy in the signal passed downstream of the photoreceptor cells [49,64,65,66]. By removing the redundant information, the contrast of objects of interest can be enhanced in the scene, leading to better object discrimination [66,67]. The rectified transient cells (RTCs) are responsible for dividing the bipolar signal coming from the LMCs into ON (positive) and OFF (negative) channels [68,69,70]. Each channel adapts to the polarity change (decreasing or increasing) in the illumination [69,70,71]. The adaptation is fast when the signal is increasing (de-polarisation) and slow when the signal is decreasing (re-polarisation) [69]. This allows for the suppression of rapid signal variations (potential background clutter) and permits only large changes (potential objects of interest) in the signal [47,72]. The elementary small target motion detectors (ESTMDs) can detect and discriminate objects of interest by correlating signals from the ON and OFF channels of the RTC stage [62,70,73]. The correlation process includes delaying the ON signal by temporal low pass filter and multiplying it by the OFF signal to detect bright objects [62,74]. For dark-object detection, the OFF signal is delayed and multiplied by the ON signal [47,70]. The MOBIVLS has been used to detect small, dim objects, such as drones, at long ranges (for more details, see [75]). The MOBIVLS processes an input image by first applying two cross-directional line scanners. Then, the output of each line scanner is split into a positive signal (the “ON” channel) and an inverted negative signal (the “OFF” channel) using a half-wave rectifier inspired by the rectified transient cells (RTCs) of the insect brain [70]. These ON–OFF channels are then passed through a multiple shift register, multiplication processing, and accumulative addition. The final detection map is generated after applying an adaptive threshold that allows the object of interest to be detected and most of the clutter suppressed [75]. Figure 2 shows a block diagram of the detection process.

The first six comparative techniques have been widely used to detect dim and small objects in infrared images. These techniques are as follows: Average Absolute Gray Difference (AAGD) [77], Improved Average Absolute Gray Difference (IAAGD) [78], High Boost Multiscale Local Contrast Measure (HB-MLCM) [79], Improved Local Contrast Measure (ILCM) [80], Multiscale Local Contrast Measure (MLCM) [81], and Multiscale Patch Contrast Measure (MPCM) [82]. In general, each of these methods computes the contrast between an object of interest and its surroundings by sliding a window around the input image vertically and horizontally, where the window centre is intended to be the object of interest. The MPCM method has the ability to detect bright and dark objects. The ILCM focuses on improving the processing speed but could negatively affect the detection performance. The HB-MLCM uses a high boost filter as a preprocessing step before applying the MLCM technique to enhance objects of interest and suppress noise and background clutter. The AAGD enhances the object of interest region by taking the difference between the average of a central window (potential object of interest region) and the average of surrounding pixels. The IAAGD solves some of the limitations of the AAGD method. It has the ability to differentiate between bright and dark objects of interest and reduce false alarms when there are sharp edges in the imagery.

The next four comparative techniques used machine-learning or neural-network techniques. Each has recently been used to detect wildlife, including koalas. The first two techniques were Faster Region Convolutional Neural Network (Faster R-CNN) [40] and You Only Look Once (YOLOv2) [41], which have been widely used in object detection, including wildlife [83,84]. Faster-RCNN is a two-stage detector where the first stage proposes regions in the imagery likely to contain objects of interest and the second stage classifies the objects of interest in these regions. YOLO is a one-stage detection network that skips the regions proposal stage and applies the detection stage directly onto the imagery. The third technique was the Template Matching Binary Mask (TMBM), which has been used to detect koalas in small areas [55]. The fourth technique was a combination of outputs (i.e., detection maps) of both Faster R-CNN and YOLOv2. In this approach, each detector processes the image sequence independently and the final output for each image is the average of the detection maps of both techniques. This idea was recently used to detect koalas in native forests [39] and eucalyptus plantations [7] and to detect Rusa deer (*Cervus timorensis*) [22]. It is called the Combined 2DCNN method, where 2DCNN refers to two deep convolutional neural networks. Combining detection maps of both models helps to reduce false alarms in the final output as the detected false alarms are less likely to be the same in both models, whereas the true detections are more likely to be the same.

#### 2.1.4. Experimental Settings

All detection techniques were programmed and run in MATLAB (R2020b). The values of the parameters used by each technique were empirically tuned to obtain maximal true positive detection rates for our datasets. Table 1 shows the parameter settings. Three distinct sets of data were used in this work: training, validation, and testing. The training data were used to retrain the pretrained networks, while the validation data were used to check the status of the training in real time. Both training and validation sets were collected from a separate location at a different time to the testing data to ensure there was no contamination between them and the test data. This separate dataset was split into 80% training and 20% validation. Once trained, all algorithms were benchmarked using the three distinct testing parameters previously mentioned: datasets A, B, and C. The neural networks (Faster R-CNN and YOLOv2) needed to be modified for koala detection. Faster R-CNN and YOLOv2 were both trained to detect koalas (one class) by fine tuning the weights of a VGG16 feature extraction network, which was pre-trained on more than a million images from the ImageNet database [85]. The fine tuning was conducted using 1239 instances of koalas from 770 images: 80% of the samples were used for training and 20% for validation. These images were recorded using the same drone and IR payload but from a different site that was not part of the three test datasets. The background of this dataset is quite similar to the three datasets used for testing. It will also be available to the public. The reason for not using the three datasets in the training is that it will lead to a bias as, for each koala in the site, some of its instances will be in the training, validation, and the test parts, which may lead to an improper experiment.

Standard data-augmentation techniques were used to improve koala detection accuracy by randomly horizontally and vertically flipping the training data to generate four versions of each sample. The raw images were normalised and stored as PNG images (bit-depth of 16) so they could be processed by the neural networks. Deep-learning techniques are usually used to detect ‘large’ objects (32×32 pixels) [42,86,87]. This is because the key features of an object of interest smaller than this are typically lost while passing through the pooling and convolutional layers of a deep neural network, where a 32×32 pixel object will be represented (at most) as 1 pixel after five processing steps by a pooling layer in the VGG16 network. This means it could easily be missed if passed through more layers [42]. Therefore, the input images were divided into tiles. Each tile was scaled up and processed so that the deep-learning algorithms (i.e., Faster R-CNN and YOLOv2) could successfully detect small objects, i.e., objects less than 7×7 pixels. This was achieved as follows: Two versions of Faster R-CNN and YOLO were implemented. First, to reduce the required processing load and match the input layer of the VGG16 network, the input image was divided into nine tiles of 224 × 224 pixels. The tiles were then processed individually by the neural networks and a detection map constructed by combining the processing outputs of all tiles. Unfortunately, due to the small size of the koalas in our dataset, neither CNN could be trained to correctly detect them. The image was therefore divided into 80 tiles of 64 × 64 pixels, and each tile scaled up to 224 × 224 pixels to match the size of the minimum input layer of the VGG16 network. These 80 tiles were then processed individually by the neural networks and a detection map constructed by combining the processing outputs of all tiles. The need to run so many versions of the neural-network classifiers on each image greatly increased the processing time for these detectors compared to the other methods. The computer used in this processing was an Alienware M15 Laptop (Dell, Miami, FL, USA) model with a Core i7-9750H 2.6 GHz CPU (Intel, Santa Clara, CA, USA), 16 GB DDR4 memory (JEDEC, Arlington, VA, USA), and NVIDIA GeForce RTX 2060 GPU (NVIDIA, Santa Clara, CA, USA).

#### 2.1.5. Performance Metrics

Equations (Equation 1) were used to evaluate the performance of all 11 comparative methods. These metrics have been widely used for the evaluation of detection techniques of objects including wildlife [1,7,23,83,88]:(1)$$\begin{array}{cc} True\;Positive\;Rate\;(TPR)\;or\;Recall & =\frac{TP}{TP+FN}\\ Precision & =\frac{TP}{TP+FP}\\ 1-Precision & =\frac{FP}{FP+TP}\\ False\;Positive\;Rate\;(FPR) & =\frac{FP}{FP+TN}\\ F1\;Score & =2\times \frac{Recall\times Precision}{Recall+Precision} \end{array}$$
where TP is the number of true positives (correct detections), FN is the number of false negatives (missing detections), FP is the number of false positives (incorrect detections), and TN is the number of true negatives (correct non-detections), where the true negative represents a patch of 12×12 pixels not part of correct detections. For all algorithms, a detection was deemed to occur correctly (TP) if the distance between the centroid of detection and the ground truth was less than or equal to six pixels (six pixels was approximately equal to 48 cm on the ground or 24 cm at the top of the trees). Two evaluation curves were computed: the receiver operating characteristic (ROC) curve and the recall vs. (1-precision) curve. These curves were computed by changing the global detection threshold from one to zero. The equal error rate (EER) and area under the ROC curve (AUROC) were also computed. The EER represents a point on the recall vs. (1-precision) curve where the recall and precision values are equal, i.e., the number of missing detections (FN) equals the number of false detections (FP). The AUROC, as its name suggests, is the area under the ROC curve. AUROC evaluates the detection rate for a technique over an FPR range rather than at a single FPR. The AUROC was computed using a linear FPR axis range of (0 and $10^{-4}$), where regions of higher false positives in the ROC curve (i.e., FPR > $10^{-4}$) are generally impractical in real-world applications.

In addition, the average detectability per koala ($Avg_{kdet}$) was computed. This has some advantages over the more basic overall/total number of koala detections. The average detectability per koala refers to the number of images that each unique koala is detected in divided by the number of images for which this koala is present. As the same koala can be observed in multiple images, the use of the average detectability per koala allows more nuanced examination of how many times the same koala could be detected by each method (relative to the total number of possible detections), as opposed to whether or not the technique was able to detect the koala only a limited number of times.

### 2.2. Results of Part 1

Figure 3(a_1–3_–c_1–3_) shows the ROC and recall vs. (1-precision) curves, as well as the AUROC and EER for all 11 comparative detection methods (AAGD, IAAGD, HB-MLCM, ILCM, MLCM, MPCM, TMBM, Faster R-CNN, YOLOv2, Combined 2DCNN, and the MOBIVLS) when tested on the datasets A, B, and C, respectively. Figure 3(d_1–3_) shows the overall results, which were computed by treating the datasets as a single entity. The MOBIVLS significantly outperformed all existing methods in terms of detection and false alarm rates, irrespective of the dataset.

Figure 3(d_3_) shows the overall percentage of area under the ROC curve (AUROC) and the overall equal error rate (EER) for the 11 comparative methods. The FPR and TPR ranges are (0–$10^{-4}$) and (0–1) for the AUROC.

Combining the Faster R-CNN and YOLOv2 [7,39] through averaging significantly improved the results. The overall AUROC and EER of the Combined 2DCNN was 46.4% and 64.8%, respectively. A 24.8% and 18.8% increase in AUROC over the individual Faster R-CNN and YOLOv2 models was recorded, along with an 18.5% and 16.5% increase for the EER. However, the AUROC for the MOBIVLS was 73.9%, a 27.5% improvement over the next best method, the Combined 2DCNN. The EER result for the MOBIVLS was 77.9%, a 13.1% improvement over the Combined 2DCNN.

Table 2 shows the overall results of the evaluation metrics described in Section 2.1.5 at FPR values of (a) $10^{-6}$ and (b) $10^{-5}$ for all of the techniques for datasets A, B, and C. It can be seen that MOBIVLS outperformed all other detection techniques tested at both levels of FPR. Individual results tables for each dataset can be seen in Appendix A (Table A1, Table A2 and Table A3).

At a FPR of $10^{-6}$, the MOBIVLS provided an F1 score of 39.8%. This compares to an F1 score of 26.3% for the next best technique, MLCM. Allowing for a FPR of $10^{-5}$, the F1 score was 76.1% (MOBIVLS) vs. 48.6% (ILCM). The MOBIVLS also showed a higher likelihood of detecting more observations of the same koala as the average detectability per koala was 67.5%, as opposed to 28.2% for the next-best method, ILCM (FPR $10^{-5}$). Overall, the MOBIVLS was able to detect 51 of the 56 koalas present, at a false alarm rate of $10^{-5}$, while the best performing alternative method (MLCM) only detected 41 and the Combined 2DCNN just 17.

In all cases, and using all metrics, the proposed MOBIVLS algorithm outperformed every other approach tested, often by a wide margin. The AUROC and EER metrics indicated that the second-best technique was the Combined 2DCNN and the third-best was the MLCM (see Figure 3(d_3_)). However, the recall, F1 score, koala count, and average detectability per koala metrics indicate the second- and third-best techniques were the MLCM and the ILCM, while the performance of the deep-learning techniques (Faster R-CNN, YOLOv2, and Combined 2DCNN) were the worst (Table A1 and Table 2). This is because the evaluation criteria in Table A1 and Table 2 were computed using a single level of FPR ($10^{-6}$ or $10^{-5}$), whereas the AUROC metric reflects the performance over a range of FPR (0–$10^{-4}$), and EER reflects the performance when the number of false negatives equals the number of false positives. As the performance of the Combined 2DCNN, the best performing supervised learning technique, surpassed the dim-object-detection techniques (at an FPR of $2.5\times 10^{-5}$, see Figure 3(d_1_)), the AUROC and EER metrics better reflect its overall performance.

Also, although deep-learning techniques provided better results in AUROC and EER metrics than dim-object-detection techniques, they require large amounts of training data, high computational power, and are relatively slow. This slow speed was primarily due to the small size of the koalas and hence the need to run multiple tiles through the detector for each image. Table 2 shows the average processing times for MATLAB implementations of the AAGD, IAAGD, HB-MLCM, ILCM, MLCM, MPCM, TMBM, Faster R-CNN, YOLOv2, and Combined 2DCNN methods. The average processing time of the unoptimised MATLAB implementation of MOBIVLS was 0.8 s per frame. Moreover, these evaluations were computed using koalas that have a detectable heat signature (as in Figure 4a,b). In other words, detections of koalas that are fully occluded (as in Figure 4c), and thus have no heat signature, were considered false alarms. This is consistent with the literature, where true detections are only computed for koalas that display a thermal signature [39].

Figure 5 shows detection maps for the eleven detection methods compared (AAGD, IAAGD, HB-MLCM, ILCM, MLCM, MPCM, TMBM, Faster R-CNN, YOLOv2, Combined 2DCNN (Faster R-CNN and YOLOv2), and MOBIVLS), applied to the image samples shown in Figure 1B. Figures of detection maps for the eleven detection methods applied to the image samples, Figure 1A,C, can be seen in Appendix A (see Figure A1 and Figure A2). For the first ten detection methods, the koalas were either completely missed, detected with low confidence, and/or had many false alarms. The Faster R-CNN and YOLOv2 performed better than the AAGD, IAAGD, HB-MLCM, ILCM, MLCM, MPCM, and TMBM techniques, and combining the results using Combined 2DCNN provided even better performance. This is because the number of false alarms decreased during the combination process. However, the Combined 2DCNN method required the koalas be detected by both the Faster R-CNN and YOLOv2 to effect higher confidence than the individual approaches. For example, as can be seen from Figure 5, where the koala in frame two was missed by YOLOv2 but detected by Faster R-CNN, the combined technique does not improve the final output. The result is a low confidence detection with many false alarms. By contrast, the MOBIVLS detected all koalas in both frames with no or very few false alarms.

It is also interesting to examine the differences and similarities between the datasets. Figure 6 shows a histogram of observed brightness temperatures for all three datasets. Datasets A and B were recorded at the same site but one hour apart (and the ground truth for both was the same, i.e., it was known that the koalas did not move between flights). Dataset C was recorded at a different site on a different day, but at approximately the same time of the day as B. The temperature distributions for sites A and B were 14.2 °C and 17.4 °C, with standard deviations of 1.35 °C and 2.64 °C, respectively. The drone, camera/payload, flight paths, forward velocity, and observation altitudes were the same for all three. However, although both the A and B datasets were recorded at the same site using the same flight paths, the images in these two datasets were not taken exactly at the same locations. Even though the A and B datasets were acquired only an hour apart, there is a slight increase in temperature (3.2 °C ± 1.3 °C) over this period. This temperature difference led to a reduction in performance. This reduction was indicated by a drop in AUROC and EER of 7–20% and 5–17%, respectively, for most methods. The exceptions were the Faster-RCNN, which fell by 35% and 24%, and the AAGD, which improved slightly (less than 1%); see Table A1 and Table A2 and Figure 3(a_1–3_,b_1–3_). This is consistent with the literature, which indicates that time of day plays an important part in the performance of automated koala detection techniques that rely on long-wave infrared (LWIR) imagery. This is because data that are not acquired early in the morning have proven challenging for koala detection techniques [20] as the relative temperature between tree canopies and the animals is lower when these surveys are conducted later in the day, and also the specular reflection from ground objects generates large numbers of false alarms [39,55,57]. However, as mentioned earlier, there is a demand for a detection method that is effective at all times of the day.

In summary, the results in the section indicated that the MOBIVLS has shown promising results when compared to the existing baseline techniques. In addition, results have shown that time of day plays an important role in degrading the performance of detection techniques. In the next sections, some analysis of real and simulation data were conducted to investigate the effects of several environmental parameters and flight settings on the detection results.

## 3. Part 2: Effects of Environmental and Flight Parameters on Koala Detection

Trees have an impact on the probability of koala detection for two main reasons. Firstly, the koala’s signature can be partially or fully occluded by canopy structures (tree stems, branches, foliage). This reduces the apparent size of the koala. Secondly, intervening canopy can (through attenuation) reduce the contrast of a koala’s heat signature relative to other objects. This reduces the ability to distinguish the koala from its background.

Several parameters relate to the first effect. These include tree structure, tree separation distance, vegetation density, koala altitude, drone altitude above canopy, and camera depression angle. Regarding the attenuating effect, there are several more complex and interdependent parameters. These include time of day, season, weather, temperature, humidity, moisture, atmospheric attenuation, emissivity of tree canopy (leaves, wood, etc.), koala emissivity, soil/ground emissivity, effect of wind (motion), and the sensitivity of the infrared camera. Moreover, these parameters are also dependent upon parameters related to the first effect. Simulating all these parameters with high fidelity would be a challenging task and is beyond the scope of this study. The goal here is to understand the broad impact of koala occlusion on the probability of detection, and to determine flight options that may eliminate, or at least reduce, such effects. Several experiments were conducted to examine this.

### 3.1. Methodology of Part 2

#### 3.1.1. Real-Data Analysis

Figure 4 shows several examples of koalas manually extracted from datasets A, B, and C, where their thermal signatures are (a) minimally attenuated/occluded; (b) somewhat attenuated/occluded, either in apparent size, contrast or both; and (c) fully occluded. The individual sub-images in (a), (b), and (c) are ’zoomed’ shots of 24×24 pixels taken from the original 640 × 512 pixel LWIR images. As can be seen, both occlusion effects discussed in the previous section (the reduction in the apparent size of koalas and the attenuation in the contrast between koalas and their surroundings) are interdependent. Thus, a koala can appear to be full size but its contrast with its surroundings is low, and vice versa. Figure 7 shows a histogram of apparent koala size and of koala vs. background intensity contrast for the three datasets and a 2D histogram for both parameters.

Figure 8 shows a sequence of images taken as the drone flew almost directly along a row of trees containing a koala. In this instance, the animal was fully occluded in frames 4 and 5; partially occluded in frames 3 and 6; and minimally (or not) occluded in frames 1, 2, 7, and 8. The main images show the koala in a set of 640 × 512 pixel images. The image in the lower right section of each picture shows a zoomed section around the koala (50×50 pixels).

Datasets A, B, and C are comprised of 2770, 2530, and 4025 frames, respectively, with koalas potentially detectable in 1482, 1356, and 412 frames, respectively (note: the potential detectability of the koalas was derived from the geometry of the drone and the spotter/ground-truth locations of the koalas). Based on the geometry alone (i.e., regardless of whether or not it was possible to manually confirm the koala detection due to occlusion), these frames contained 2336, 2175, and 412 koala detections. There were 25 unique koalas in datasets A and B and 6 in dataset C. In 40%, 42%, and 36% of these locations, based on manual verification, koala heat signatures were completely blocked and could not be visually identified in datasets A, B, and C, respectively. These ratios were computed by checking the koala locations in the frames that have koalas and declaring full occlusion when no thermal signature was visible (as in Figure 4c). The average overall full occlusion ratio for the three datasets was 40%, i.e., if a koala was present in 100 images, its heat signature could be expected to be detectable in only 60 of these images.

Figure 9 shows the attenuation probability for the horizontal (across-track) field of view (HFOV) and vertical (along-track) field of view (VFOV), respectively, for the three datasets and the average overall results. It can be seen that koalas are more likely to be fully occluded near the centre of the image as the average probability of attenuation increased by 27% and 13% with respect to the HFOV and VFOV, respectively. This indicates koalas are less likely to be detected when they are on trees directly beneath the drone. However, this was based on our datasets that employed drone collection strategies using a camera in nadir. To generalise the statement, more data need to be collected at different sites, times of the day, seasons, and weather conditions, with the camera at different angles and in plantations as well as in native forest.

#### 3.1.2. Simulation Analysis

A series of simple simulation experiments were conducted to analyse the effect of different flight configurations on the probability of detecting koalas. In these experiments, different simulated tree structures, tree separation distances, and koala heights were used with a variety of camera tilt angles and flight altitudes. A description of the tree’s structure, koala altitude, and drone-camera flight settings were simulated as follows:Tree’s structure: Trees in mature eucalyptus plantations are mainly composed of straight stems with an average height of 30 m, diameter of 30 cm, and uniform spacing between trees of approximately 3 m. Figure 10 shows two images taken at different eucalyptus plantations. The structure of trees in native forests is very different to these, being far more variable.Figure 11 shows a graphical representation of four simulated tree structures. These images show (a) tree stems; (b) stems and branches; (c) stems, branches, and foliage of 1 m diameter; and (d) stems, branches, and heavier foliage of 2 m diameter. Each has a progressively greater occluding effect on koala detection. In the simulation, tree stems are represented as a uniform lattice of 30 m high vertical cylinders of 30 cm diameter, and branches are represented by four cylinders of 1 m length and 15 cm diameter at an elevation angle of 45°, oriented as a cross at the top of each tree. Foliage is represented as a spherical blob, diameter 1 m for light foliage and 2 m for heavy foliage. Any ray travelling from the drone to the koala that penetrates or grazes a tree structure was considered to fully attenuate the koala’s signature. In other words, the model depicted a binary occlusion model, with no attenuation or spatial diminution represented (known as the Beer–Lambert law).In reality, the structure of tree canopy and branches is highly complex and depends on many (interdependent) environmental factors. It is thus challenging to realistically simulate all such elements, and such an undertaking is beyond the scope of this paper. However, whilst somewhat unrepresentative of the physical world, the model suffices as a rough indicator in terms of the impact that trees, branches, and foliage have on the occlusion of koala signatures.Koala altitude: Koalas were modelled as spheres, radius 25 cm, at three heights (see Figure 11). They were placed at altitudes of 25 m, 15 m, and 0 m. These altitudes represent near the top of a 30 m tree, which is where koalas are often found (in the foliage), just below the main foliage (near the middle of the tree), and on the ground next to a tree.Drone-camera flight settings: The simulated flight altitude was varied from 40 m up to 100 m over trees of 30 m height, i.e., 10 to 50 m above the canopy. The camera depression angle was varied from 0 (horizontal) to 90° (nadir) in 10° increments. The camera had a 50 × 50° field of view with a 640 × 640 pixel image resolution.The trigonometric relationship between drone altitude (fh), camera depression angle (da), and koala altitude (ka) and the koala size in the captured images is depicted in Figure 12 and Equations (Equation 2) and (Equation 3).
(2)$$dca=\frac{fh-ka}{cos(90-da+\theta )}$$
(3)$$Koala\;size\;\left(in\;pixels\right)=\frac{ImageDimension\;(in\;pixels)\times KoalaSize\;(in\;m)}{2\times dca\;\left(in\;m\right)\times tan\left(\frac{FOV}{2}\right)}$$
where dca refers to distance between camera and koala (in m), fh refers to flight height (in m), ka refers to koala height (in m), da refers to depression angle (in degrees), and θ refers to camera incident angle of the field of view (in degrees) (See Figure 12).

**Figure 10 sensors-24-07048-f010:**
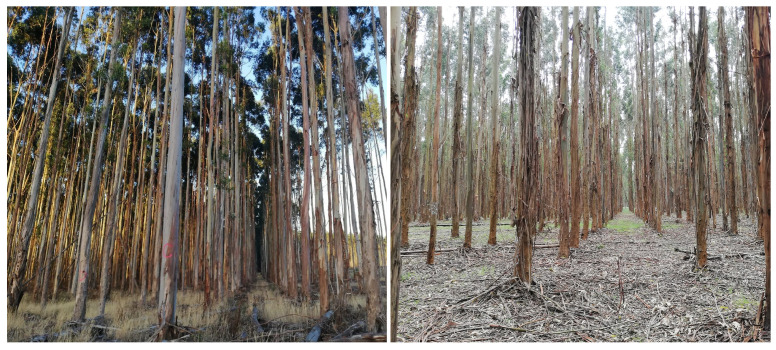
Two images taken at different eucalyptus plantations where tree stems are straight and spacing between trees is uniform.

**Figure 11 sensors-24-07048-f011:**
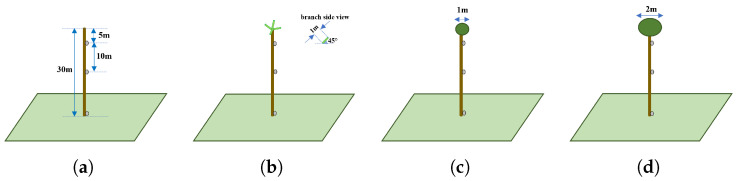
Four types of simulated tree structures that enabled the occluding effects of tree stems, branches, and foliage to be examined. The tree structure comprised (**a**) tree stems, (**b**) branches, (**c**) foliage of 1 m diameter (light canopy), and (**d**) foliage of 2 m diameter (heavy canopy). Tree stems were represented as a uniform lattice of vertical cylinders, 30 m high and with a diameter of 30 cm. Branches were represented by four cylinders of 1 m length and 15 cm diameter oriented at 45° (elevation angle) to the horizontal and distributed as a cross at the top of each tree. Foliage was represented as a spherical blob, with a diameter of 1 m for light canopy and 2 m for heavy canopy. The koala was simulated as a sphere of radius 25 cm at three different altitudes: 25 m, 15 m, 0 m (Note: This is only an illustrative sketch so the dimensions are not to scale).

**Figure 12 sensors-24-07048-f012:**
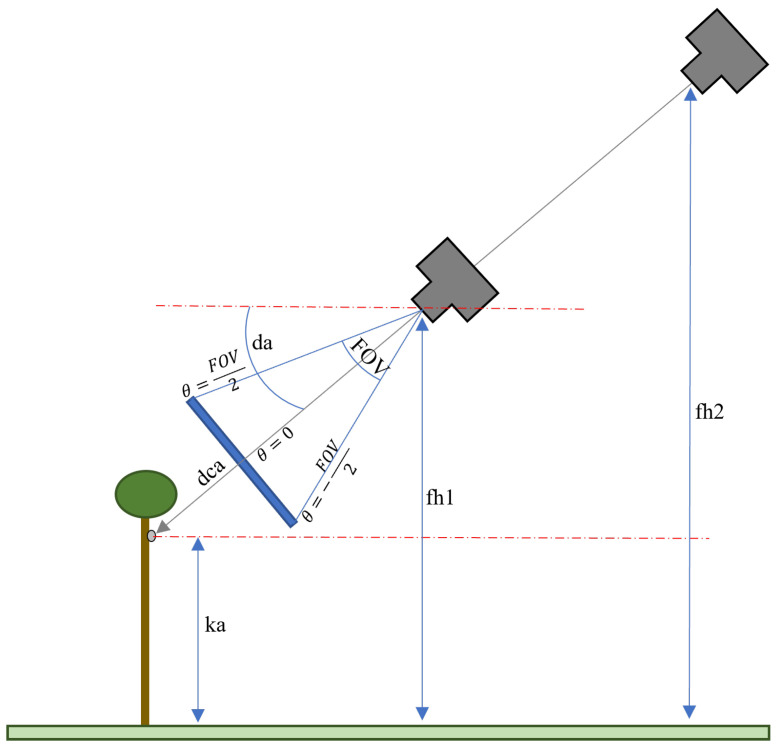
A sketch to demonstrate the mathematical relation between the flight height (fh), camera depression angle (da), koala altitude (ka), camera incident angle of the field of view (FOV), and the koala size in the captured images.

### 3.2. Results of Part 2

The simulation was encoded using the Unity Engine 5 software [89]. The simulation environment was composed of a square grid of 9 × 9 trees, with a simulated koala attached to the central tree. For each experiment, the camera overflew the tree grid in a lawn-mower pattern, with a 2 m separation between adjacent transects (the approximate angular discrimination between transects was 2.9–1.9° for 40 m and 60 m flight altitudes, respectively). The transects were set close enough to ensure that the environment was captured from different perspectives of the camera along-track and across-track field of views. The along- and across-track extrema for the flight transects was such that the tree containing the simulated koala was visible in all of the images. The camera captured an image from the scene every 2 m in the along-track direction.

To avoid any bias, each simulation was repeated eight times, with the koala oriented on a different side of the central tree each time (north, north-east, east, south-east, south, south-west, west, and north-west). For each experiment, the probability of koala detection (true positive rate) was then computed as per Equation (Equation 1). The detector used to extract the koala from images was a simple threshold filter. That is, ray paths were considered to travel from the centre of the drone to each koala pixel, and any that intersected a tree structure (stem, branch, or foliage) were considered to block the path to the koala area in that angle of the field of view entirely. In other words, the model represented binary occlusion (no intensity signature attenuation), with only a spatial diminution represented.

The first simulation investigated the effect of tree separation distance on the probability of koala detection. Figure 13 shows the probability of detection using 1 m, 3 m, and 5 m separation distances for the four different tree structures (stems only, stems with branches, and stems with 1 m and 2 m foliage, as detailed previously) as a function of camera depression angle range (0–90°), three koala altitudes (25 m, 15 m, and 0 m on a tree height of 30 m), and a drone height of 40 m, i.e., 10 m above the canopy. Despite the simplicity of the simulation, the results show several trends that should be considered when planning for aerial koala surveys.

First, in the absence of foliage, koalas are less occluded when they are close to the tree tops. This is self-evident as the occlusion is caused by the stems alone. Second, the tree separation distance had a significant impact on koala occlusion, and this effect increased when koalas were closer to the ground. In other words, koalas are more likely to be missed due to occlusion in high-density plantations or forests. Third, the camera tilt angle should be at nadir (in Figure 13, nadir is a depression angle of 90°), except in high tree density areas with heavy foliage, where the camera tilt angle should be angled in the range 40–60° to view beneath the ’impenetrable’ canopy. In Figure 13a, as the heavy foliage was simply represented as a sphere of 2 m diameter that fully occludes anything behind it (binary occlusion), and the tree separation distance was 1 m, any koala close to the top of the trees (but below the canopy) could not be detected when the camera was at nadir. However, in Figure 13b,c, when the koala was at lower altitudes (15 m and 0 m, respectively) and they are located far below the foliage, their detectability improves relative to Figure 13a. In all cases in Figure 13a–c, the detection was optimal when tilting the camera in the range of 40–60° for the scenario of heavy foliage and 1 m tree separation distance.

In Figure 13d,e,g,h, it can be seen that the probability of detection drops around a depression angle of 40°, especially in the scenario of heavy foliage (seen more clearly in Figure 13d). This drop in detectability is an artefact of the simple geometric model used in the simulation, and the reason is again related to the simplicity of the model used and the geometry between the camera tilt angle and the location of koalas relative to the canopy structure and tree separation distance. The simplicity of the model precludes nuanced findings being reported, as these are likely artefacts of the model used. Nevertheless, key messages may be drawn: (a) for readily penetrable foliage scenarios orienting the camera at nadir offers the highest detection probability (lowest occlusion), unless the trees are packed very tightly together, (b) the impact of tree stems alone (no foliage) can have a dramatic impact on koala occlusion, even for koalas high up in the trees, and (c) aside from when trees are very densely packed (1 m separation distance), the effect of light foliage (branches and 1 m foliage) relative to tree stems alone is relatively modest.

The second simulation experiment investigated the effect of flight height on the probability of koala detection/occlusion. Figure 14 shows the probability of detection for drone flights 10 m, 30 m, and 50 m above the canopy, for a 3 m tree stem separation distance and the four different foliage structures used previously. Once again, the detection probability is plotted as a function of camera depression angle for three koala altitudes (25 m, 15 m, and 0 m) on trees of 30 m, and the detector used to extract the koala from captured images was (again) a simple threshold filter. Whilst the probability of detection due to occlusion did not change much as a function of drone altitude, it should be noted that the size of a koala’s signature decreased as the range increased. Figure 15 provides an estimate of koala size in pixels, ignoring any occlusion effect of trees, etc. (note: $25^{2}$ pixels means 25 × 25 pixels, not 5 × 5 pixels). In this figure, the koala is represented as a sphere of 25 cm radius; drone altitude (fh) varied between 40 m and 100 m; camera depression angle (da) from 0 to 90°; and koala altitude (ka) of 25 m, 15 m, and 0 m on a 30 m tree. The results shown are for the centre of the camera’s field of view, θ (θ = 0°). The trigonometric relationship between these factors is depicted in Figure 12 and Equations (Equation 2) and (Equation 3). The effect of signature diminution had a more significant impact on the probability of detection in real datasets than the simulated datasets as not every detection technique can detect koala signatures ‘cleanly’, i.e., if the signatures are small (i.e., less than 5×5 pixels) and detected, they are likely accompanied by a large number of false alarms (not simulated in this model). Note also that techniques such as CNN-based detectors cannot generally detect such small signatures, whilst for others, such as MOBIVLS, small objects are not so problematic.

Figure 15a shows that, for koalas close to the tree tops, their signature is more than 5 × 5 pixels for flight altitudes up to 70 m above the canopy and camera depression angles of 80–90° and that, as the depression angle decreases, the flight height required to keep the koala at or above 5 × 5 pixels also decreases in a roughly linear fashion, i.e., for a depression angle of 20° the maximum altitude is about 20 m. For koalas on or near the ground (Figure 15c), the flight altitude required to keep the koala signature at or above 5 × 5 pixels decreases to 40 m above canopy, and once again, as the depression angle decreases, the flight altitude required to keep the koala size at or above 5 × 5 pixels decreases, albeit this time to 20 m above the canopy for a depression angle of 50°.

Figure 16 shows how koala signature size varies as a function of camera zenith/depression angle. It can be seen that the apparent koala size varies as a function of image location and is related to the camera angle, an effect that is quite pronounced for shallow (near horizontal) angles (0–40°).

To obtain an insight into how foliage might attenuate the koala signature, simplistic calculations were conducted where the rate of attenuation, α, is assumed to have a linear relationship to the integrated path length of a ray as it passes through foliage blocking the direct line of sight between the camera and koala (note: this relationship is often known as the Beer–Lambert law). Although the assumption is not a perfect representation of physical reality, it is a reasonable assumption that offers insight into the effect. Figure 17 shows the results of this simple attenuation experiment. The camera height was 40 m (10 m above the canopy); koala altitude 25 m; foliage represented by sphere of radius 1 m; and tree separation distances of 1 m, 3 m, and 5 m, as shown in Figure 17a, b, and c, respectively.

Figure 17d, e, and f depict plots for relative attenuation for Figure 17a, b, and c, respectively. The *x*-axis represents the horizontal distance between the camera and the koala. The values in the *x*- and *y*-axes are shown as a percentage of the foliage radius (R) so that they are more physically meaningful. Attenuation values may be computed by multiplying the value on the y-axis by an appropriate value of α. As the separation distance between trees decreased, the attenuation effect of foliage increased, and vice versa, and heavier foliage had a greater attenuating impact on the koala signature. The artefacts/spikes, especially in Figure 17d, are related to the simplicity of the geometric model used in the simulation. This simplicity of the model precludes nuanced findings being reported, as these are likely artefacts of the model used.

Nevertheless, key conclusions may be drawn: (a) there is an inverse relationship between the separation distance between trees and the attenuation effect of foliage on koala heat signatures, and (b) aside from when trees are very densely packed (1 m separation distance), koala signatures tend to be attenuated by canopy structure when they are directly beneath the drone and less likely to be attenuated when they are away. This is consistent with the findings from real data (see Figure 9), where it was noted that koala signatures were more likely to be occluded (less detectable) when they are close to the centre of the image than on its sides. However, this was based on our datasets and drone collection strategies (camera in nadir), and to generalise the statement, more data need to be collected and processed from different sites (plantations and native forests), times of the day, seasons, and weather conditions.

## 4. Discussion

This paper examined the performance of the newly proposed novel Multiscale Object Bio-Inspired Vision Line Scanner technique (MOBIVLS) [75] versus ten existing detection techniques on three LWIR datasets of koalas in eucalyptus plantations. The ten techniques are composed of six LWIR small-object-detection techniques widely used in the literature to detect dim and small objects [77,78,79,80,81,82], four computer-vision and CNN-based techniques that have previously been used to detect different objects (including koalas) [7,39,40,41,55]. The results show that the MOBIVLS method significantly outperforms all other methods tested in this study (including the deep-learning techniques), and that the approach was able to detect koalas in high-clutter environments with low false alarm rates. This was despite the observations being made during the day as opposed to in the early morning, where the temperate differential makes detection simpler. It is also noted that MOBIVLS does not need to be pre-trained [75]. Results could be further improved by using a tracker or data association between sequences of frames [7,23,39,55], and this will be investigated in future work.

Time of day is known to have a significant impact on the performance of automated koala detection techniques using LWIR imagery due to the diurnal effects of solar heating. It therefore needs to be considered when planning field trials. Preliminary results show that a one-hour time difference between two datasets recorded at the same site (using the same drone, camera payload, flight path, altitude, etc.) increased the mean recorded brightness temperature by 3 °C ± 1 °C. This small increase in temperature appears to have degraded the performance of the automatic detection techniques, as measured by AUROC and EER, by 7–20% and 5–17%, respectively. High summer temperatures reduce the contrast between koalas and their surroundings. Therefore, it increases the difficulty of detecting them. Moreover, as the environment becomes heated, this will also likely increase false detections as more heat reflections will be falsely detected as koalas. This is shown when comparing the results of datasets A and B, which comprises images of the same site but with a one-hour time difference, and a corresponding temperate change between them. It is noted, however, that one sample of data is insufficient to draw conclusions, and to thoroughly investigate the effect of temperature/time of day, field trials need to be carried out over several days at different months of the year (i.e., different seasons) and at multiple sites.

The results also indicate that occlusion by tree stems and canopy structure has a significant impact on the potential detection of koalas, with the animals fully occluded in up to 40% of images known to have koalas. However, this finding was based on a limited number of datasets, and to generalise the statement, more datasets need to be examined.

Simple simulation experiments were conducted to examine the effect of different flight configurations on the effects of occlusion on the probability of koala detection. Despite the simplicity of the model employed, the experiments offer some useful insights for planning aerial koala surveys. Firstly, the position of the koala on the tree has a high impact on koala occlusion, with detection probability decreasing when koalas are closer to the ground. Secondly, camera tilt angle should be at nadir (depression angle of 90°), or at least in the range of 80–90°, except in cases of high-density trees with heavy foliage. In these cases, tilting the camera to 40–60° should provide better results. In the literature, although there are studies that used nadir [39,55] and oblique imagery [7,55] to detect koalas, there is no quantitative assessment or comparison as to which mode provides better detection results. This simulation therefore provides a step towards understanding the effect of camera tilt angle on koala detection.

Our simulation experiments are consistent with the extant literature regarding the effect of flight altitude on koala detection, finding that the probability of detection increases when flying at lower altitudes, i.e., closer to the tree tops [55,90]. This is to be expected as flying at lower altitudes ensures koalas occupy more pixels in the captured images, and this reduces the likelihood of a detector missing them. However, flying at low altitudes is not always possible (or safe) as trees are usually not of uniform height and terrain is not always even, especially in natural forests. In general, the flight altitude is not recommended to be more than 40 m above the canopy as the mathematical calculations suggest that the size of a koala in the captured images are expected to be smaller than 8×8 and 5×5 pixels (for the camera configuration used in this study) when they are at 25 m and 0 m altitudes, respectively, and this assumes there is no occlusion. In reality, the effects of foliage will reduce the effective size of the koala, and other environmental effects (weather, temperature, humidity, etc.) may diminish the signatures even further.

## 5. Conclusions

This paper presents contributions in the area of object detection, with a particular emphasis on detecting wildlife in low contrast, high clutter environments. A biologically inspired object-detection algorithm, known as the Multiscale Object Bio-Inspired Vision Line Scanner, or MOBIVLS, is presented and tested on three datasets. The MOBIVLS can process single images and does not require high frame rate temporal information, as is the case for other biologically inspired vision techniques.

The main focus of part 1 of this paper is on koala detection using long wave infrared imagery. Despite their small size and low levels of contrast against their surroundings in high-clutter environments, the MOBIVLS outperformed all state-of-the-art detection techniques to which it was compared and for all datasets that were examined. It achieved 74% area under the receiver operating characteristic curve (AUROC), a measure of average detection probability. This represented a 27% improvement over the next best method when used to detect koalas in aerial infrared images. MOBIVLS dynamically adapts to a range of environments without the need for training.

In part 2, in addition to applying MOBIVLS to infrared aerial datasets containing koalas, several other issues associated with koala detection in infrared aerial images were examined using a combination of simulation and real data. These included the nature of koala occlusion by tree structures, the effects of plantation properties on the effects of occlusion (and hence the probability of koala detection), and the effect of different flight parameters. The main findings indicate that koala occlusion by tree structure can have a significant impact on the potential detection of koalas, with koalas fully occluded in up to 40% of images in which they would otherwise be expected to be detected. However, this finding was based on a limited number of datasets, and to generalise the statement, more data need to be collected and processed for different sites, stand properties, times of day, season, and weather conditions. For the best results, based on the simple geometric model used to examine the matter, the drone’s camera should generally be set to nadir view (90°), or at least in the range of 80–90°. In certain circumstances, such as very high-density trees (1 m spacing) or heavy foliage, setting a camera tilt angle in the range 40–60° and flying closer to the tree tops can achieve better results than nadir.

In the future, it is planned to experiment with the MOBIVLS approach for the detection of more varied wildlife in forestry areas and also to investigate how the work could be adapted for the detection of objects of interest in urban environments.

## Figures and Tables

**Figure 1 sensors-24-07048-f001:**
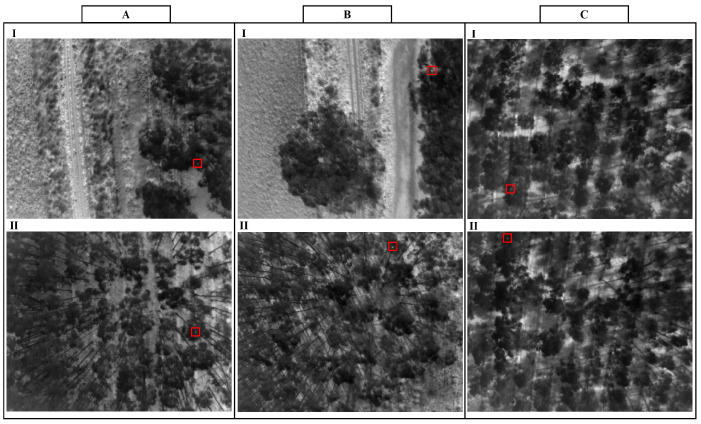
Two image samples from each dataset (**A**–**C**). Each image contains a koala in a eucalyptus plantation, highlighted by a red box.

**Figure 2 sensors-24-07048-f002:**
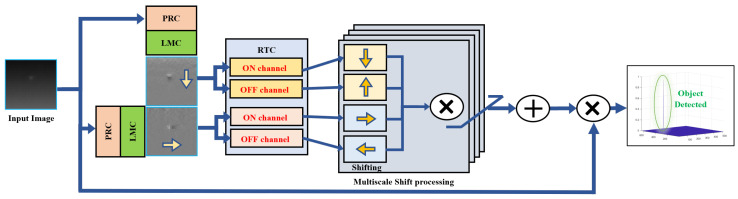
Block diagram of the multiscale object of bio-inspired vision line scanners (MOBIVLS). The × symbol refers to a Hadamard product [76], while the + symbol refers to normal array addition. Images output from the LMC stages show that the object contrast has been enhanced in the direction of the scanning. PRC, LMC, and RTC stands for photoreceptor cell, lamina monopolar cell, and rectified transient cell, inspired from insects vision pathway.

**Figure 3 sensors-24-07048-f003:**
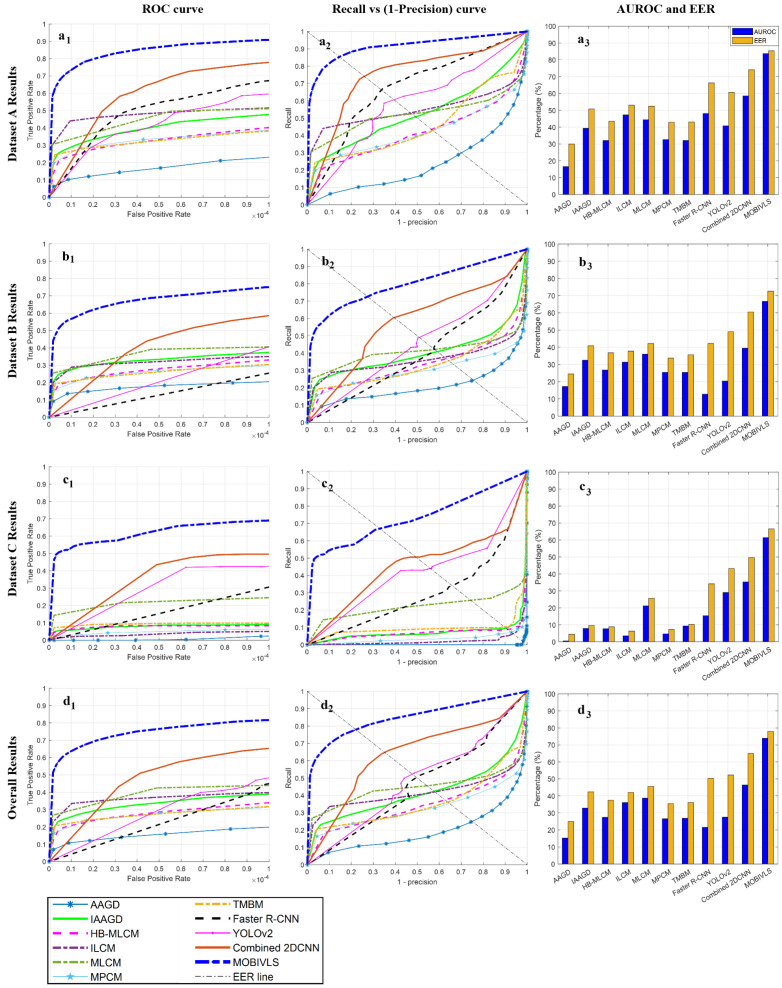
Evaluation curves for 11 comparative koala detection techniques (AAGD, IAAGD, HB-MLCM, ILCM, MLCM, MPCM, TMBM, Faster R-CNN, YOLOv2, Combined 2DCNN, and the MOBIVLS): (**a_1_**–**d_1_**) show the receiver operating characteristic (ROC) curves (TPR vs. FPR); (**a_2_**–**d_2_**) show the recall vs. (1-precision) curves; and (**a_3_**–**d_3_**) show the AUROC and EER percentages. The FPR range over which the AUROC calculations were computed was (0–$10^{-4}$), while TPR range used was (0–1). The uppermost three rows of Figures show the results from datasets A–C, respectively, with the last row showing the overall (average) results. In all cases, the proposed MOBIVLS algorithm outperformed all of the other approaches tested.

**Figure 4 sensors-24-07048-f004:**
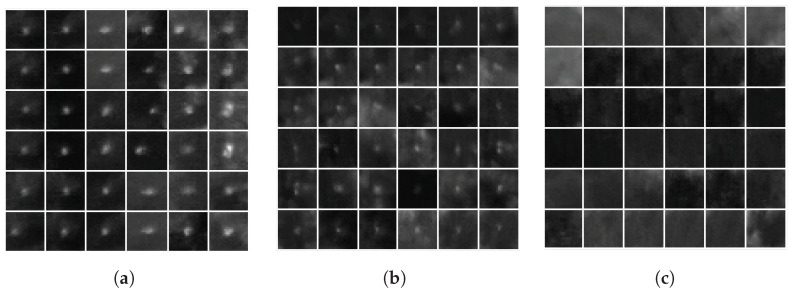
Instances of different koalas from datasets A, B, and C, where (**a**) shows instances of minimally attenuated koala heat signatures, (**b**) shows instances of somewhat attenuated koala heat signatures (either in apparent size, contrast, or both), and (**c**) shows instances of koala heat signatures fully attenuated/occluded. The individual sub-images in (**a**–**c**) are ‘zoomed’ 24×24 pixel patches taken from the original 640 × 512 pixel LWIR images. Koalas are approximately central in each image.

**Figure 5 sensors-24-07048-f005:**
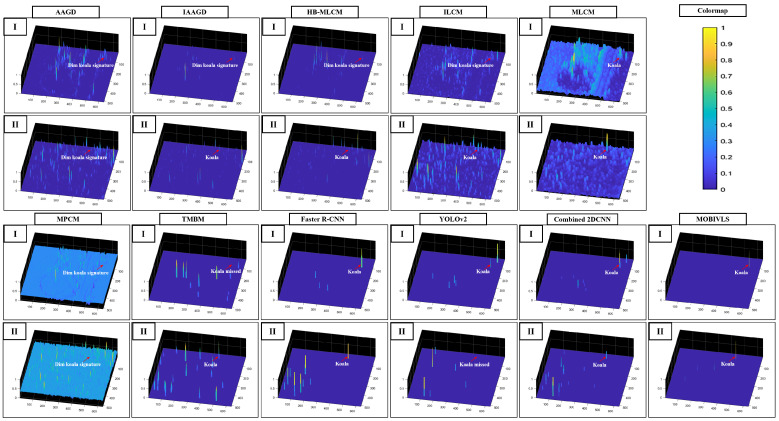
Detection maps generated by 11 comparative object detection methods applied to infrared images. For each detection method, images I and II show processed samples for the raw data shown in Figure 1B. In each case, more false detections were generated, and/or the koala response was a lower contrast than for the MOBIVLS method (best viewed in digital format).

**Figure 6 sensors-24-07048-f006:**
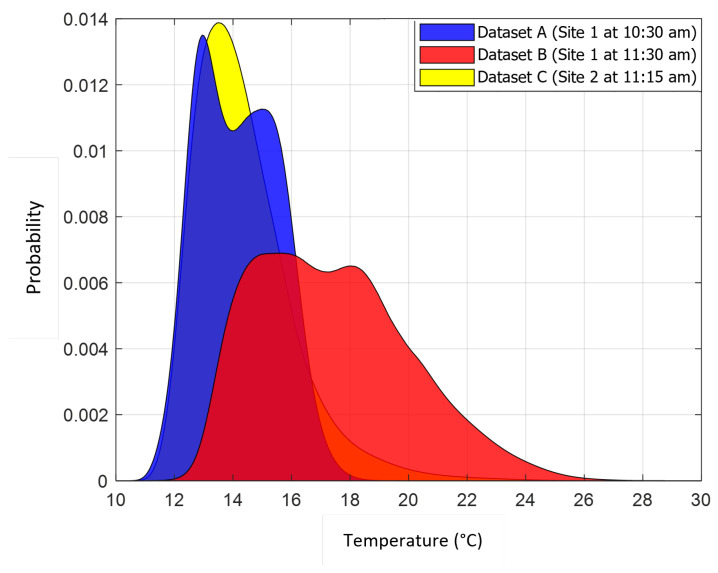
Temperature distributions for the three datasets. Datasets A and B were recorded at the same site one hour apart; dataset C was recorded at a similar time of day to B but at a different location. The *x*-axis represents the brightness temperature values captured by the IR camera. The *y*-axis represents the probability of occurrence (note: the summation of frequency for all values of each dataset is equal to one).

**Figure 7 sensors-24-07048-f007:**
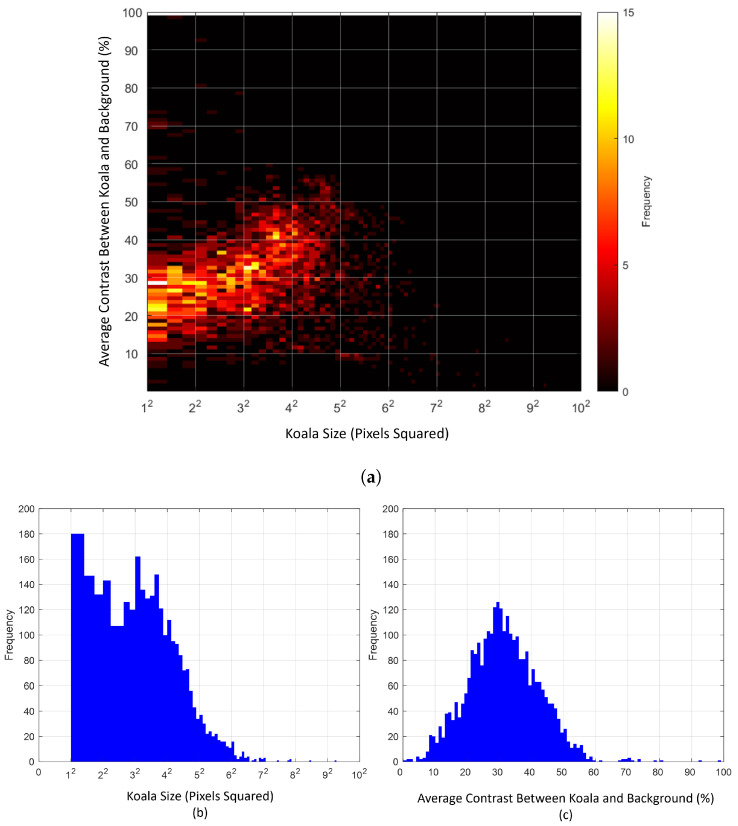
(**a**) shows a 2D histogram of koala sizes versus their average contrast with respect to their surrounding background for all instances of koala detections in datasets A, B, and C. (**b**,**c**) show histograms of koala sizes and their average contrast with their surrounding background, respectively. The data are drawn from all 3250 detections in datasets A, B, and C. The average contrast between koala and background was computed by differencing the average intensity of koala pixels and the average of their surrounding background pixels within the 24×24 pixel patches immediately around the koalas.

**Figure 8 sensors-24-07048-f008:**
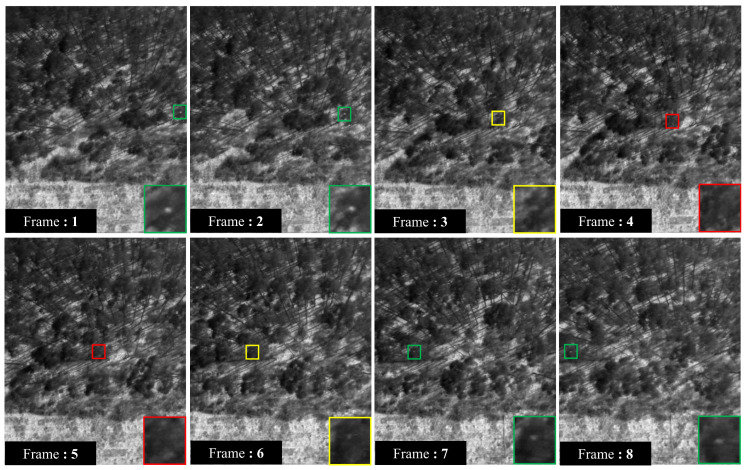
Sequence of images showing the same koala for different occlusion cases. The red, yellow, and green bounding boxes show cases when the koala is fully, partially, and not obscured by tree canopies or trunks. The UAV flew from left to right so the position of the koala in the image sequence appears to move from right to left. Zoomed in regions around the koala (50×50 pixels) are shown in the lower right of each image for enhanced clarity.

**Figure 9 sensors-24-07048-f009:**
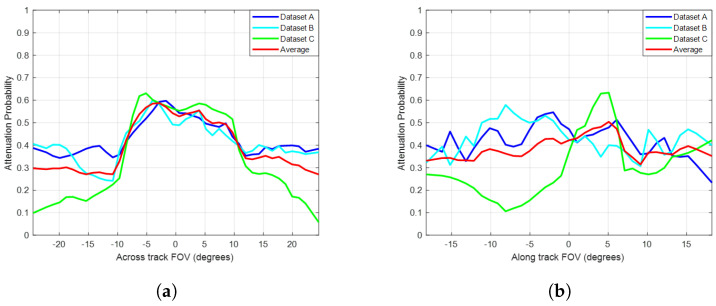
The attenuation probability with respect to the camera horizontal (across-track) field of view (HFOV) (**a**) and vertical (along-track) field of view (VFOV) (**b**) for the three datasets and the average overall results. The *y*-axis represents the attenuation probability computed based on the accumulation of koala observations within images, and the *x*-axis represents the across-track or along-track field of view, where the HFOV was 50° (−24.5° to 24.5°) and the VFOV was 37.5° (−18.25° to 18.25°).

**Figure 13 sensors-24-07048-f013:**
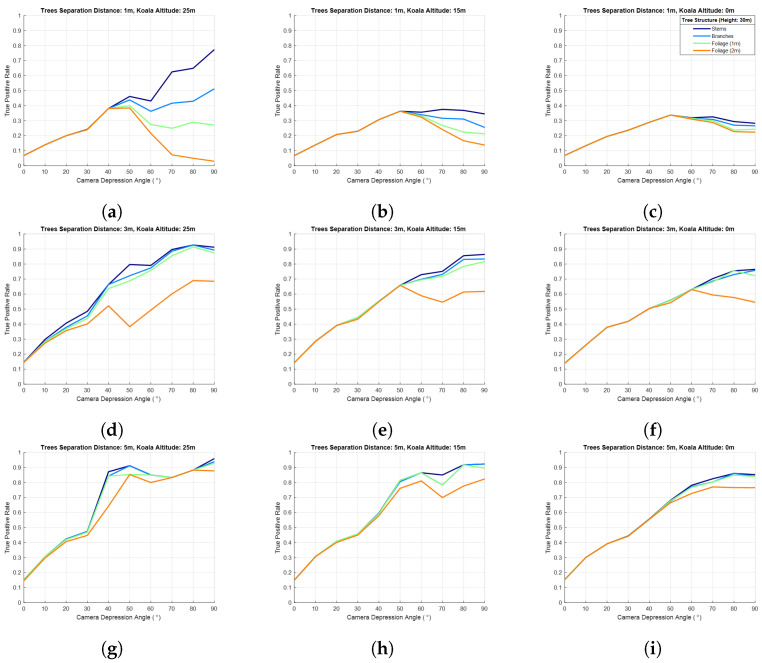
The true positive rates (probability of detection, *y*-axis) for a range of simulation experiments using 1 m, 3 m, and 5 m tree separation distances (upper (**a**–**c**), middle (**d**–**f**), and bottom (**g**–**i**) rows, respectively) for four types of tree structure (tree stems alone (dark blue lines), tree stems with branches (light blue lines), tree stems with foliage of 1 m diameter (green lines), and tree stems with heavier foliage of 2 m diameter (orange lines)), camera depression angle range (0–90°) (*x*-axis), and three koala altitudes (25 m (left column), 15 m (centre column) and 0 m (right hand column)). Tree height was 30 m and drone height was 40 m, i.e., 10 m above the canopy. Simulated koalas were always placed adjacent to a tree stem.

**Figure 14 sensors-24-07048-f014:**
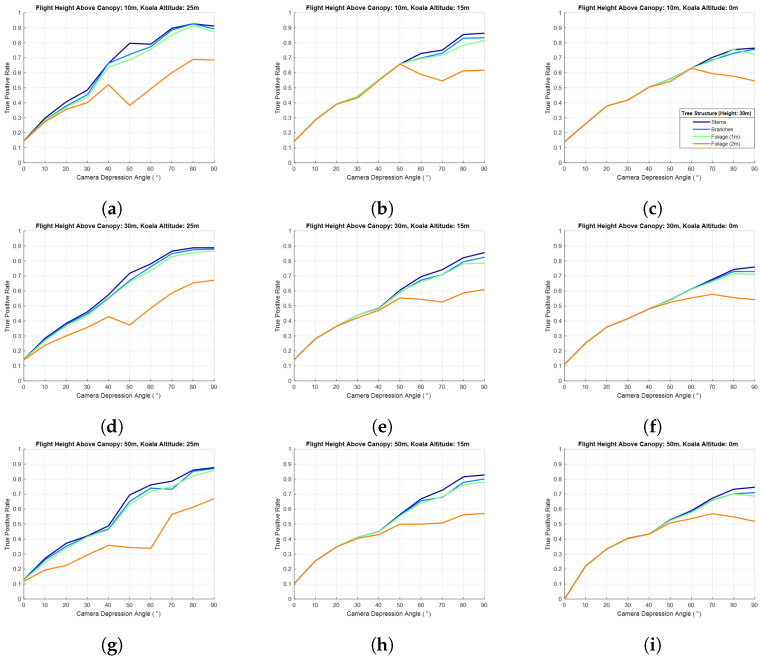
True positive rates (probability of detection) of the simulation experiments that make use of 3 m tree separation distances for four types of tree structure (tree stems only, branches, foliage of 1 m diameter, and heavier foliage of 2 m diameter), camera depression angle range (0–90°), three koala altitudes (left hand column: 25 m, centre column: 15 m, right hand column: 0 m) on a tree of 30 m height and flight heights of (upper row: 10 m (**a**–**c**), centre row: 30 m (**d**–**f**), lower row: 50 m (**g**–**i**)) above canopy (i.e., 40 m, 60 m, 80 m above ground).

**Figure 15 sensors-24-07048-f015:**
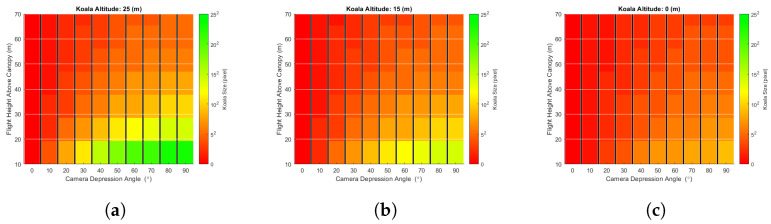
An estimate of koala size in pixels with respect to flight height range (40–100 m), camera depression angle range (0–90°), and centre of the camera (field of view (θ): 0°) for different koala altitudes: (**a**) 25 m, (**b**) 15 m, (**c**) 0 m. The equations used in these calculations (see Equations (Equation 2) and (Equation 3)) provide an estimate of koala size (in pixels) in an ideal case and ignore any occlusion effect of trees or the surrounding environment.

**Figure 16 sensors-24-07048-f016:**
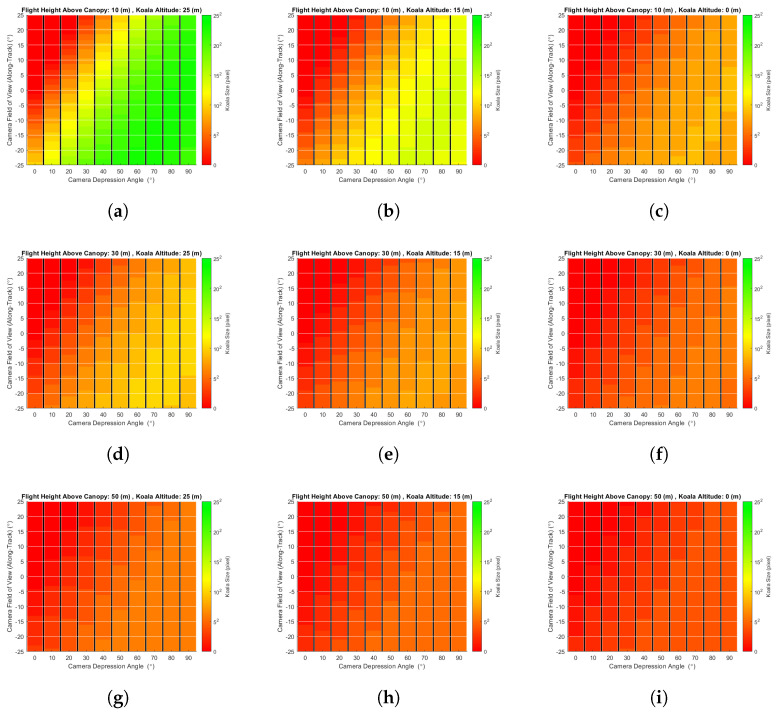
Koala size (in pixels) as a function of incident camera angles and field of view for different koala and flight altitudes. The estimated koala sizes were computed with respect to flight height above the canopy (upper row (**a**–**c**): 10 m, centre row (**d**–**f**): 30 m, lower row (**g**–**i**): 50 m), camera depression angle range (0–90°), koala altitude (left hand column: 25 m, centre column: 15 m, right hand column: 0 m), and camera field of view, θ = −25° to 25°. The equations used in these calculations (Equations (Equation 2) and (Equation 3)) provide an estimate of koala size for ideal cases only and ignores the occluding effects of trees and their environment.

**Figure 17 sensors-24-07048-f017:**
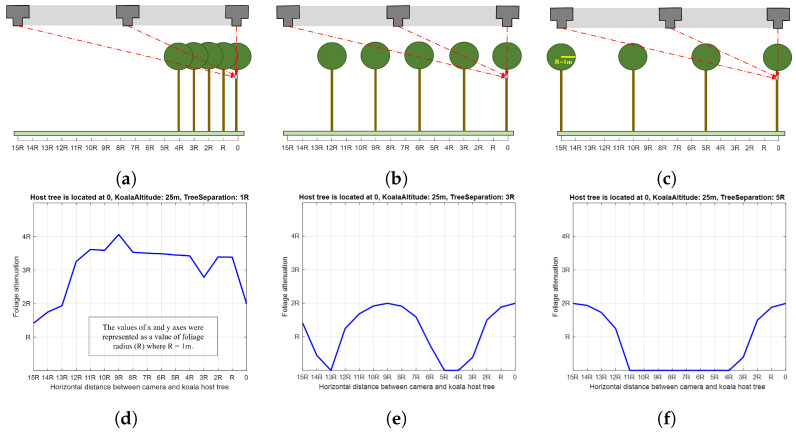
Simulation-based estimate of the amount of foliage obstructing the direct line between a camera and a koala. The heavier the foliage, the more a koala signature was attenuated, and vice versa. Therefore, foliage attenuation (*y*-axis) may be used as a surrogate for foliage density (Note: Although this is not an accurate representation of real-world data, it does give a rough indication of the attenuating effect of foliage on koala signatures). The camera height was fixed at 40 m (10 m above the canopy); the koala was at a height of 25 m; the foliage was represented as a sphere of radius (R) 1 m; and the tree separation distance was 1 m, 3 m, and 5 m, as shown in (**a**–**c**), respectively (note: these are only illustrative sketches so the dimensions are not to scale). (**d**–**f**) show the amount of foliage attenuation that obstructs a direct line between a camera and the koala for each figures directly above it. The *x*-axis represents the horizontal distance between the camera and the koala host tree.

**Table 1 sensors-24-07048-t001:** Parameter settings of different methods.

No.	Methods	Parameter Settings
1	**AAGD** [77]	Inner window scales: 3, 5, 7, 9, 11 pixels
2	**IAAGD** [78]	Outer window scales: 21, 21, 21
3	**HB-MLCM** [79]	21 pixels
4	**ILCM** [80]	Window scale: 10 pixels, step size: 1 pixel
5	**MLCM** [81]	Object window scales: 3, 5, 7, 9, 11
6	**MPCM** [82]	
7	**TMBM** [55]	Template Processing: subtracting
		constant (C): 0.2, threshold (T): 0.6
		Scoring threshold: 0.8, pixel intensity
		threshold: 0.1
8	**Faster R-CNN**	Backbone: VGG16, batch size: 8,
	[7,39,40]	epoch: 100, initial learning rate: $10^{-5}$
9	**YOLOv2**	Backbone: VGG16, batch size: 32,
	[7,39,41]	epoch: 100, initial learning rate: $10^{-5}$
10	**Combined 2DCNN**	2DCNN are Faster R-CNN and YOLOv2,
	[7,39]	parameter settings: same as No. 8–9
11	**MOBIVLS** [75]	Object scales 3–11 pixels

**Table 2 sensors-24-07048-t002:** The overall results of the three datasets, which were computed by treating the datasets as a single entity. Several performance metrics were computed for different object-detection techniques at (a) FPR of $10^{-6}$ and (b) $10^{-5}$. The total number of unique koalas was 56. The best result of each metric is highlighted by an underline and bold style. The second-best result is indicated by bold style only. The proposed MOBIVLS algorithm performed better than all of the other techniques against all of the metrics used. The column ‘Time’ represents the processing time in seconds per frame.

No.	Methods	Recall (%)		F1 Score (%)		Koala Count		Avg_*kdet*_ (%)		Time
		a	b	a	b	a	b	a	b	
1	**AAGD** [77]	3.6	11	6.6	19.1	9	24	3.6	11.1	0.06
2	**IAAGD** [78]	9.4	25.9	15.6	40	17	35	8.7	23.7	0.18
3	**HB-MLCM** [79]	6.7	20.9	11.8	33.5	16	28	6.5	19.7	0.06
4	**ILCM** [80]	14.3	33.3	23.1	48.6	24	36	11.9	28.2	0.11
5	**MLCM** [81]	16.8	29.6	26.3	44.9	**26**	**41**	**16**	28.1	0.37
6	**MPCM** [82]	8	20.4	13.6	33	16	32	7.5	18.9	0.25
7	**TMBM** [55]	8.2	22.3	13.4	35.5	14	35	7.6	20.6	0.053
8	**Faster R-CNN**	0.4	4.3	0.5	5	3	13	0.5	4.6	14
	[7,39,40]									
9	**YOLOv2**	0.6	6	0.8	7.6	3	17	0.7	7.2	3.5
	[7,39,41]									
10	**Combined 2DCNN**	1.4	13.7	1.8	17.7	3	17	1.6	15.7	17.5
	[7,39]									
11	**MOBIVLS** [75]	30.2	63.7	39.8	76.1	**32**	**51**	32.1	67.5	0.8

## Data Availability

Data are contained within the article.

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
