# Peer review of "Evaluation of Automated Object-Detection Algorithms for Koala Detection in Infrared Aerial Imagery"

_sensors, 2024, doi:10.3390/s24217048_

Round 1

Reviewer 1 Report

Comments and Suggestions for Authors

The study offers valuable insights into automated koala detection techniques in eucalyptus plantations by introducing the novel Multi-Scale Bionic Visual Line Scanner technique (MOBIVLS) and comparing it to ten existing methods. The study design appears robust, and the findings have significant potential to enhance wildlife monitoring and conservation efforts. However, there are certain areas, particularly in terms of logical coherence, that could benefit from further refinement. I believe that addressing the following points could improve the overall quality of the manuscript:

1.     The introduction section could benefit from enhanced logical coherence. For example, while the second paragraph provides valuable references on the application of drones in animal monitoring, it might be more effective if it also highlighted the advantages and limitations of this approach. Furthermore, the transition to discussing thermal imaging studies in the third paragraph feels somewhat abrupt. A smoother connection could improve the overall flow of the introduction.

2.     In the methods section, lines 162-164 mention that the drones flight altitude was approximately 60 meters, which is 35 meters above the tree canopy. However, in lines 540-543 of section 4.2.3, it indicates that the flight altitude varied from 40 to 80 meters. Additionally, section 4.2.4 highlights the influence of flight altitude on monitoring results. This discrepancy raises a question about the consistency of the information presented in the methods section. It might be helpful to clarify this aspect to enhance the overall clarity of the manuscript.

3.     It might enhance the readability of the paper and better emphasize its key points if Tables 2 to 5 were merged, or if Tables 2 to 4 were included in the supporting materials. This could help streamline the presentation of information for the readers.

4.     It could be beneficial to consider combining Figures 3, 4, and 5, or partially placing them in the supporting materials. Since these figures primarily illustrate detection maps from 11 comparative object detection methods applied to infrared images, with the main distinction being the study location, a more streamlined presentation might enhance clarity for the readers.

5.     In the results section, it might be beneficial to relocate the descriptions of Figures 12 and 16, as well as Equations 2 and 3, to the Methods section. This adjustment could enhance clarity regarding the context and methodology prior to discussing the results.

6.     The article would benefit from the inclusion of a discussion section. While the manuscript addresses existing detection techniques, adding a quantitative comparison of their performance against MOBIVLS could enhance the analysis. This inclusion would effectively highlight the strengths of the new method and provide readers with a clearer understanding of its practical advantages.

Comments on the Quality of English Language

Enhancing the logical flow of the English language in the article could significantly improve its overall coherence.

Author Response

Please see the attachment. Response to Reviewer 1 Comments section

Reviewer 2 Report

Comments and Suggestions for Authors

This is a useful study that applies novel insect-inspired biological vision systems to detect koalas in IR imagery.  The results are very encouraging, showing that the MOBIVLS outperforms more traditional IR enhancement procedures and deep learning approaches in detecting koalas in a very complex wooded environment.

In the Introduction I would have liked to see a short statement about why this study is important in the context of koala conservation in Australia.  The rate of decline of koalas, as related to habitat alteration and/or destruction, and climate change effects.  Why it is important for studies such as this to try to enumerate koala populations to guide conservation efforts?

On Methods, it seems strange that the flight missions were conducted so late in the morning when koala TIR signal becomes attenuated by rising background temperature.

I feel that a diagram is needed to clarify the mechanism of the MOBIVLS system.  I came away from the methods section without a clear basic understanding of how MOBIVLS works and how it is applied to IR video imagery.

I also did not understand the training, validation and testing procedures for the deep learning YOLO and Faster-R-CNN networks.  It seems the study re-trained on models derived elsewhere.  We know nothing of these ‘external’ models.  And was the re-training done with box or point annotation?  Would not it have been better to use the study’s own A, B and C datasets to run a traditional 70%:20%:10% training:validation:testing procedure for determining deep learning performance, since this picks up the unique and individual background of each study area?

The assessment of tree canopy effects, although based on simulation, is useful and interesting.  It provides a template for modelling the detection of animals in other wooded environments.

Some more specific comments are listed below:

Line 154.  Why were the data acquisition flight performed so late in the day?  10.30 – 11.30?  Was it to give the ground teams time to find the koalas?  Should be stated.

Line 207.  The MOBIVLS process is highly technical, and needs to be explained more clearly for readers who are using AI to detect animals on imagery.  It is hard to understand how the two cross-directional (presumably orthogonal) line-scanner signals operate through moving images to generate a detection.  A diagram is essential to show this, and must be included.

Line 277.  This first paragraph describing the process of Faster R-CNN and YOLO training needs to be better explained.  There is a pre-trained model presented, where does this originate from?  Is this pre-trained model also derived from IR imagery of koalas?  As I understand, this pre-trained model is fine-tuned using imagery from another test area using the same UAV with IR payload to avoid ‘contamination’ of the primary A, B and C datasets, presumably meaning that there entire primary A, B and C datasets were for testing only.  Was there any discussion about using the A, B, and C datasets for the training, divided by (e.g) 70%:20%:10% into training: validation; testing? Or were there not enough positive koala images to do this?

Line 324.  What is the unit of measure (ie the count) for a True Negative?  It means that model did not detect anything, when nothing was there, but is the 'unit of detection' the whole individual image or tile?

Line 399.  “In other words, detections of koalas that are fully occluded (as in Figure 7(c)), and thus have no heat signature, were considered false alarms. This is consistent with the literature where true detections are only computed for koalas that display a thermal signature [37].”.   A True Detection (True Positive) presumably relates to detecting a koala in the same location as identified by the ground teams.  What happens if the ground truthing indicates it is there, but the model said it is not there (because there are leaves and branches, and the signal is attenuated).  This is not a false alarm (a False Positive).  This is a False Negative.  Is this built into the metrics?

Line 428:  This issue of time of day is critical, but is rather brushed over.  Whilst the effects of increasing background temperature is described at the end of this paper, it should be indicated much earlier in methods.  These flights were done in the late morning of the Australian springtime.  Surely, if this had been done earlier in the day, with lower ground and ambient air temperature, the detections would have been more pronounced.

 Figure 8.  “The average contrast between koala and background was computed by differencing the average intensity of koala pixels and the average of their surrounding background pixels within the 24 _ 24 pixel patches immediately around the koalas”.  How is this calculated on the scale of 1-100%?  A koala with the same intensity (eg value 16) as background (eg value 16) is the ratio of the two, ie 100%.

A reference to the below article might strengthen the case for oblique IR imagery for animal detection.:

Lethbridge, M., Stead, M. and Wells, C. (2019) ‘Estimating kangaroo density by aerial survey: a comparison of thermal cameras with human observers’, Wildlife Research. doi: doi.org/10.1071/WR18122

Comments on the Quality of English Language

The quality of English of this article is reasonably satisfactory.  However, there is much repetition, eg "It is important to note that..." and "koalas (the subject of this study)".  There are many instances where the narrative requires a new paragraph to avoid confusion and introduce clarity.  Some suggestions to improve English are shown below:

Line 10.  Do you mean to say Biologically vision systems…” .  Shouldn’t it be “Biological vision systems..”   ?

Line 12.  A very long sentence…  “Therefore, this paper introduces a biologically-inspired small object detection 13 algorithm and evaluates its performance against ten other detection techniques, including both 14 image processing and neural network based approaches using three datasets containing koalas in 15 eucalyptus plantations.”  This should be split or reworded, for example: “Therefore, this paper introduces a biologically-inspired detection algorithm to locate koalas in three Eucalyptus test areas, and evaluates its performance against ten other detection techniques, including both image processing and neural network based approaches”.

 Line 18.  “The analysis of simulated and real data shows that koala occlusion  by tree stems and canopy can have a significant impact on the potential detection of koalas, with  koalas being fully occluded in up to 40% of images in which koalas they were known to be present”.

 The ability to accurately enumerate identify wildlife populations is fundamental to many conservation and natural resource management programs [1,2]. For instance, accurately determining a species population size and is essential for drawing up effective plans to protect endangered species [3], monitoring the movement and behaviour of migratory animals [4], managing populations of invasive species within eradication programs [5,6], optimising protection of wildlife during natural disasters.

 However, detecting wildlife is a challenging task. Wildlife moves, is often camouflaged against its background [7] and the environmental background is often “cluttered” consisting of a complex interplay of elements. Despite the difficulty of detection, land managers require accurate data to achieve their objectives without negatively impacting wildlife. However, traditional wildlife surveying methods, including diurnal searches, nocturnal spotlighting [8,9], detection dogs and radio or satellite collars, camera traps [9] and hand-held thermal imagers [10], all have limitations. In addition, they are often time-consuming, labour intensive, expensive, and some are hazardous [1].

 Line 55.  Automatic detection through deep learning techniques can identify critical features extracted from imagery that are necessary to detect objects of interest [3033]. Very wordy.  Try…

Automated deep learning techniques can identify and extract critical features from imagery to detect objects of interest.

 Line 66.  Try to bring these sentences together for clarity.  “This necessity to use object features for classification can be a significant problem in datasets with very small items of interest (occupying fewer than 10 pixels) where occlusion is common. Thus further reducing the visible size of the objects [sic].”

 Line 101.  Repetition. No need to repeat .. (the species of interest in this paper)”…

 Line 161. Better… “It should be noted that …”

Lines 166 and 167.  Should be the same paragraph.

Line 167:  So at 10Hz, and the indicated number of frames, flights were approximately 4-6 minutes?  Should be stated.

Then Line 169.  New paragraph.. “To create ground-truth..”

Line 172…   why is there a backslash ?   \

Line 180.  “Important to mention” is redundant text. (also see earlier line 178)

Line 209.  Insect inspired, repetition

Line 279.  Table 1… should not be in brackets

Author Response

Please see the attachment. Response to Reviewer 2 Comments Section

Reviewer 3 Report

Comments and Suggestions for Authors

1.     Given that the three datasets share identical acquisition conditions and image dimensions, differing only in acquisition times, it is worth considering merging them into a single dataset.

2.     The limited size of the three LWIR datasets may hinder a comprehensive assessment of MOBIVLS's generalization capabilities.  It is standard practice to employ a larger and more diverse collection of datasets, encompassing data from various regions and seasons, to ensure robust evaluation.  Furthermore, clarifying the practical implications of monitoring koalas specifically within the chosen region and time frame would enhance the study's context.

3.     The simulation experiment's simplified modeling of leaves and canopy may not accurately represent real-world scenarios, potentially impacting the reliability of the simulation results.  Comparing MOBIVLS against end-to-end models could offer valuable insights into its strengths and limitations.

4.     Finally, the font size in some subfigures appears too small and requires adjustment for improved readability.

Author Response

Please see the attachment. Response to Reviewer 3 Comments Section

Reviewer 4 Report

Comments and Suggestions for Authors

This paper introduces a biologically-inspired small object detection algorithm and evaluates its performance. It also examines koala occlusion by canopy cover in these plantations. Finally, it describes a new accurate ground truth dataset of aerial high dynamic range infrared imagery containing koala heat signatures. Firstly, the structure of the three parts is not appropriately balanced; the most important algorithm section is too brief while the latter two parts are excessively lengthy. Secondly, the paper’s grammar and logic are not smooth.

Here are some specific issues:

1. The title is too long; it is recommended to shorten it and highlight the key points. The biologically-inspired detection algorithm mentioned in the title occupies too little space in the paper, and the factors analyzed that affect koala detection are not limited to just canopy cover but also include flight altitude and temperature.

2. In the abstract, “The use of infrared cameras and drones has shown promising results, irrespective of whether the detection was carried out by human observers or automated algorithms. Consequently, automated detection is particularly desirable as it eliminates several key issues related to human factors such as fatigue and tedium, while also enhancing safety and surveying speeds.” The causal relationship is not valid.

3. It is recommended to summarize the conclusions of sections 3, 4, and 5 in the introduction; otherwise, it is too vague.

4. The paper should clearly distinguish between small object detection and occluded object detection, as these are two different concepts.

5. The section “Koala Detection Results” lacks a concluding summary paragraph.

6. Some related works are missing. [1] Cognition-Driven Structural Prior for Instance-Dependent Label Transition Matrix Estimation 

Comments on the Quality of English Language

NAN

Author Response

Please see the attachment. Response to Reviewer 4 Comments Section

Round 2

Reviewer 1 Report

Comments and Suggestions for Authors

Thank you for the revisions made to the paper, which have contributed to some improvement in its logical coherence. However, the inclusion of a table of contents in the revised manuscript may not be an essential component of the paper’s structure. Please consider whether it is necessary to retain this section. Additionally, the overall structure of the paper has not been adjusted as per the suggestions. The methods section remains in the fourth part, and there is no dedicated discussion section, which may affect the overall logical flow of the paper. I kindly encourage the author to reconsider whether further adjustments are needed to enhance the paper's structure.

Author Response

Please see the attachment. Response to Reviewer1 section

Reviewer 4 Report

Comments and Suggestions for Authors

Authors answers all my concerns.
Some suggestion related works may be added.[1] Part-Aware Correlation Networks for Few-shot Learning

Comments on the Quality of English Language

NAN

Author Response

Please see the attachment. Response to Reviewer4 section
